# Short-Term Survival and Postoperative Complications Rates in Horses Undergoing Colic Surgery: A Multicentre Study

**DOI:** 10.3390/ani13061107

**Published:** 2023-03-20

**Authors:** Alessandro Spadari, Rodolfo Gialletti, Marco Gandini, Emanuela Valle, Anna Cerullo, Damiano Cavallini, Alice Bertoletti, Riccardo Rinnovati, Giulia Forni, Nicola Scilimati, Gessica Giusto

**Affiliations:** 1Department of Veterinary Medical Sciences, University of Bologna, 40136 Bologna, Italy; 2Veterinary Teaching Hospital, Department of Veterinary Medicine, 06123 Perugia, Italy; 3Department of Veterinary Sciences, University of Turin, 10095 Grugliasco, Italy

**Keywords:** horse, colic, emergency laparotomy, short-term survival, predictive factors

## Abstract

**Simple Summary:**

Colic surgery, despite the improvements in recent years, is not without risks and still has a high risk of death compared with other procedures. Although about 90% of cases of colic in horses resolve spontaneously or with medical treatment, the remaining 10% can be fatal if not treated surgically. Furthermore, postoperative complications can have important welfare and economic consequences. Studies of predictive prognosis indices, incidence of postoperative complications, and survival rates in different geographical areas may not be comparable. Consequently, there is a need to perform a study that investigates the Italian population of animals subjected to colic surgery. A total of 451 horses were included. The short-term survival rate was 68.5% for all the horses that underwent colic surgery and 80% of the horses surviving anaesthesia. Several risk factors were evaluated and age, body condition score (BCS), packed cell volume (PCV) and total plasma protein (TPP) before and after surgery, amount of reflux, type of disease, type of lesion, duration of surgery and surgeon’s experience, and amount of intra- and postoperative fluids administered affected the outcome. The multivariate analysis revealed that PCV at arrival, TPP after surgery, and BCS had the highest predictive power. This is the first multicentre study in Italy and the results of this study could help surgeons choose the best treatment and clearly communicate risks to referring veterinarians and owners. Further prospective studies should be conducted to confirm the effect of predictive indices considered in this study on short-term survival.

**Abstract:**

The occurrence of colic could be influenced by the characteristics of a population, geographical area, and feeding management. The aim of this study was to report the short-term postoperative complications and survival rates and to identify factors that might affect the outcome of horses that underwent colic surgery in three Italian surgical referral centres. Data of horses subjected to colic surgery in three referral centres (2018–2021) were analysed. Comparisons of the outcomes were performed using a Mann–Whitney or a Chi square test. Areas under the receiver operating characteristic (ROC) curve and multivariable logistic regression analysis were used for parameters that were significant in the previous univariate analysis. The goodness-of-fit of the model was assessed using the Akike information criterion (AIC). Significance was defined as *p* < 0.05, and odds ratios and 95% confidence intervals were calculated as percentages. A total of 451 horses were included. The survival rate was 68.5% of all of the horses that underwent colic surgery and 80% of the horses surviving anaesthesia. Age, BCS, PCV and TPP before and after surgery, amount of reflux, type of disease, type of lesion, duration of surgery, surgeon’s experience, and amount of intra- and postoperative fluids administered influenced the probability of short-term survival. The multivariate analysis revealed that PCV at arrival, TPP after surgery, and BCS had the highest predictive power. This is the first multicentre study in Italy. The results of this study may help surgeons to inform owners regarding the prognosis of colic surgery.

## 1. Introduction

Although about 90% of cases of colic in horses resolve spontaneously or with medical treatment, the remaining 10% of colic cases can be fatal if not treated surgically [1]. Surgery, despite improvements in recent years, is not without risks [2] and still has a high risk of death compared with other procedures, both for surgery and anaesthesia-related issues [3]. Furthermore, postoperative complications can have important welfare and economic consequences. Thus, it is essential to know the factors that affect prognosis in order to help owners, veterinarians, and surgeons to make optimal decisions about treatments [4]. 

Several studies have already been published about the complications and survival rates of horses undergoing colic surgery [4,5,6,7,8,9,10,11]. These studies have reported that the short-term survival rate could be affected by several factors, such as the type and severity of disease or the treatment applied [4,5,6,7,8,9,10,11,12,13]. To the best of our knowledge, no multicentre studies have been conducted in Italy describing predictive indices of prognosis and short-term survival rates in horses undergoing colic surgery. 

The incidence of gastrointestinal diseases and the survival rate of patients undergoing colic surgeries may depend on several factors. Each equine population has its own intrinsic or extrinsic characteristics in terms of diet, management, or weather conditions that could influence the onset of colic [14]. In fact, depending on the geographical area, exposure to specific risk factors, such as the types of forage and climatic conditions, is different [15,16,17]. Seasons are recognized as a risk factor for colic, but the results of previous studies are contradictory, with some studies reporting hot months and others cold months as predisposing for the occurrence of symptoms [16]. For these reasons, studies on predictive prognosis indices, incidence of postoperative complications, and survival rates in different areas may not be comparable. Consequently, there is a need to perform a study to investigate the Italian population of animals undergoing colic surgery.

The aim of this study was to analyse part of the Italian population of horses referred for colic to three referral centres in Northern and central Italy. The analysis was aimed at identifying risk factors related to short-term survival. The identification of variables associated with the outcome will allow the surgeon to choose the best treatment for each clinical case and in clearly communicating the risks associated with it to the owners.

## 2. Materials and Methods

This was a retrospective cohort study evaluating horses undergoing colic surgery in three different surgical centres in Italy between 2018 and 2021.

Case records and variables

Medical records of horses undergoing colic surgery in the Veterinary Teaching Hospitals (VTHs) of the Universities of Bologna, Perugia, and Turin (Italy), between 2018 and 2021, were analysed in a retrospective manner. Horse signalment, clinical and surgical data, short-term postoperative clinical features, treatment, first surgeon, complications, and outcome were retrieved. No exclusion criteria were used, and incomplete records were considered acceptable.

Definitions

Delta PCV and Delta TPP were defined as the difference between PCV and TPP, respectively, at arrival and soon after recovery from surgery.

Short-term postoperative complications were defined as any deviation from the normal postoperative course [18,19] before discharge. Short-term survival was defined as survival until discharge from the hospital.

The outcome was categorized as positive or negative if horses survived to discharge or not.

Statistical analysis

Statistical analysis was performed using statistical software (JMP Pro 16JMP Statistical Discovery LLC, SAS Institute srl, Milano, Italy). Descriptive statistics were generated (mean ± S.D., median and range) for continuous data (count and percentage) and for categorical data of all horses that underwent colic surgery. Normality was assessed using the Shapiro–Wilk test for continuous variables. The differences between horses surviving and not surviving to discharge were assessed using a Mann–Whitney test for continuous variables and a chi-squared test for categorical variables. Any association between the surgeon’s number of surgeries per year and outcome was evaluated using Spearman’s correlation. 

Areas under the receiver operating characteristic (ROC) curve were used for parameters that were significant in the univariate analysis to evaluate discrimination and to provide specificity and sensitivity prediction values. Classes were then created according to ROC discriminant values to integrate in the previous model as categorical variables. 

Variables that showed statistical significance in previously applied models were selected for a multivariable logistic regression analysis. The goodness-of-fit of the model was assessed using Akike information criterion (AIC). Significance was set at *p* < 0.05. Odds ratios (OR) and 95% confidence intervals (95% CI) were calculated for the categorical data. 

## 3. Results

Case details

A total of 451 horses were included in the study, of which 76/451 (16.8%) were referred to the VTH of Bologna, 162/451 (35.9%) to the VTH of Perugia, and 213/451 (47.2%) to the VTH of Turin. Two hundred and one horses were female (44.5%), 183 were geldings (40.5%), and 67 were stallions (14.8%). The median age was 12 years (range = 0.2–32), median weight was 491 kg (range = 30–740), and median body condition score (BCS) was 6 (range = 2–9). Most of the horses were Warmblood (53.9%), followed by hot blood (Thoroughbreds and Arabs, 16.4%), pony (9.5%), American Saddlebred (9.1%), Trotter (6.9%), cold blood (1.8%), and mule (0.2%). In 2.2% of cases the breed was not recorded.

Clinical features upon arrival

Details of the clinical parameters at arrival are reported in Table 1 for horses with strangulating lesions and in Table 2 for horses with non-strangulating lesions.

Most of the animals were last seen healthy 24 h or less before being admitted to the hospital (22.7%; n = 91). Data on when the animal was last seen healthy for the last time before being referred to a hospital are shown in Table 3.

Surgical features

Details of horses that died during surgery are reported in Figure 1. Of the 33 horses euthanized during surgery, 14 were referred to the VTH of Bologna, 12 to the VTH of Perugia, and 7 to the VTH of Turin. In 7/33 horses, reasons for intraoperative euthanasia were reported (2 horses were euthanized because of SIRS and 5 for the involvement of too much intestine). 

Table 4 shows the different sections of the intestinal tract involved, while Table 5 shows the incidence of specific diagnoses.

Strangulating lesions were recorded in 39.9% of cases. Strangulating lesions involving the large intestine were recorded in 70/254 (27.5%) cases and the small intestine in 110/191 (55.8%) cases. The type of anastomosis is reported in Table 6.

The median duration of surgery in horses recovered from anaesthesia was 1.58 h (range: 1–5.17; n = 384). The median volume of pre- and intraoperative fluid therapy was 10 L (range: 0.3–118; n = 316).

Postoperative clinical features

Data concerning postoperative clinical variables are reported in Table 7.

Short-term postoperative complications

Eighteen out of 386 (4.6%) horses underwent relaparotomy within the same hospitalization period for the recurrence of clinical signs of colic. Data regarding horses that underwent relaparotomy are reported in Figure 2.

Eighteen horses were submitted to relaparatomy in the same hospitalization period. Adhesions were the cause of colic necessitating relaparotomy in 5 cases (27.7%); intestinal impaction in 3 (16.6%); and LCV, peritonitis, or POI in 2 cases (11.1%) each. Acute abdominal herniation, RDD, anastomotic obstruction, or intussusception were the cause of colic necessitating laparotomy in one horse each.

Single or multiple episodes of pyrexia (>38.6 °C) were the most common postoperative complication and were recorded in 27.7% of 386 horses recovered from anaesthesia. A complete list and % of postoperative complications are reported in Table 8.

Short-term survival and predictive indicators

The short-term survival rate was 68.5% for all 451 horses that underwent colic surgery and 80% for 386 horses that recovered from anaesthesia. Prognostic predictive indices were evaluated only in hospitalized horses, excluding animals that underwent intraoperative euthanasia. The use of data from subjects judged to be inoperable, or that underwent euthanasia for economic reasons, may represent a confounding factor when identifying risk factors for the prognosis of horses. The characteristics of these subjects were considered separately from those that survived surgery. 

There was no significant association between sex, weight, breed, and survival, while age influenced the outcome. Based on an evaluation of the ROC curve, horses aged > 14 years were 2.3 times as likely to have a negative outcome than horses between 10 and 14 years of age (*p* = 0.0276). Both in horses < or >14 y.o. the most common lesions were RDD, LCV, large colon impaction, and EFE. Most (89.4%) cases of pedunculated lipoma were found in horses > 14 years. Nephrosplenic entrapment was found mostly in horses < 14 years (68.2%).

BCS also influenced the outcome and a ROC curve was evaluated. Horses with a BCS between 4 and 6 and >6 had a 2.83- and 3-times higher probability of a positive outcome than horses with BCS < 4 (*p* = 0.013; *p* < 0.01). 

PCV, TPP, blood lactate and volume of reflux appear to be good predictors of outcome. A cut-off was identified using the ROC curve for each variable. Horses with PCV > 50% (*p* < 0.0003) and TPP < 5.7 (*p* = 0.0034) and >7.4 (*p* = 0.0207) upon arrival had a poorer prognosis than horses with other values. Blood lactate < 1.2 mmol/L was associated with a higher probability of a positive outcome than blood lactate > 6.6 mmol/L (*p* = 0.05). The presence of reflux did not affect the outcome, but if the volume of reflux was related to bodyweight, it was observed that horses with a volume of reflux less than 0.018 L/kg BW had a better outcome (*p* = 0.0109). The referral time did not affect the outcome. 

Horses with pathology of viscera other than the intestine (e.g., spleen, liver) had a 7.3-times greater probability of a negative outcome than horses with intestinal disease (*p* = 0.0316). In addition, the diagnosis influenced the outcome. Horses with LCV, pedunculated lipoma, and small intestinal volvulus had the worst outcome. Details of the outcomes for the most common diagnoses are reported in Table 9.

Horses with non-strangulating lesions had a 2.18 times higher probability of a positive outcome than horses with strangulating lesions. In horses that did not undergo resection and anastomosis, the probability of a positive outcome was 2.3 times higher than in horses that underwent intestinal resection and anastomosis (*p* < 0.01). The type of anastomosis did not affect the outcome. Surgery lasting more than 2 h and 24 min was more likely to result in a negative outcome (*p* < 0.0001). Surgeon experience, in terms of the number of horses operated per year of experience, affected the outcome. A mean number of horses operated per year of experience >18 was correlated with a better short-term outcome (r = 0.83). Details relating to surgeons’ experience are reported in Table 10. 

Furthermore, a volume of crystalloids > 44 mL/kg administered pre- and intraoperatively (i.e., between arrival and recovery from anaesthesia) was associated with a poorer prognosis (*p* = 0.0208). No hypertonic or colloids were used. 

PCV after surgery also influenced the outcome. Using ROC curves, two PCV values were identified as cut-offs: a PCV between 26 and 43% was associated with a higher probability of a positive outcome (*p* = 0.0105). Furthermore, the outcome was better if the change in PCV (delta PCV) was between –8 and 0 than if it was less than –8 (*p* = 0.0156). The TPP value after surgery and delta TPP also affected the outcome, as a TPP value after surgery below 4.5 g/dL was associated with a worse outcome (*p* = 0.0229) and a decrease of > 2.2 g/dL was associated with a higher probability of a negative outcome (*p* = 0.0089). 

In multi-variable logistic regression models, PCV at arrival, TPP after surgery, and BCS were the factors with the highest predictive power, as reported in Table 11.

## 4. Discussion

To the best of our knowledge, this is the first multicentre study in Italy reporting short-term postoperative complications, survival rate, and risk factors that affect the short-term outcome of horses subjected to colic surgery. Although previous similar studies have identified multiple risk factors related to equine colic, these findings may not be the same when evaluated in populations with different characteristics. The short-term survival rate of horses in our study (80%) was a little lower than in some other studies [4,6,7] and higher than in others [5,20,21].

To evaluate the risk factors that could influence short-term survival, only hospitalized horses were included because the criteria for defining an inoperable subject are subjective and thus can vary greatly between centres and surgeons. Furthermore, the data of animals that underwent intraoperative euthanasia for economic reasons were excluded because these could be confounding factors. 

In the univariate model in our study, several predictive indices associated with short-term survival were identified. As these risk factors could be influenced by extrinsic and intrinsic characteristics of equine populations [14], we focused on a part of the Italian population referred to three different centres. Although the equine population considered in this study was different to the populations in previous studies, we obtained similar findings, demonstrating that sex, weight, and breed were not associated with outcome, while age; cardiovascular parameters at arrival such as PCV, blood lactate, and TPP values; cardiovascular parameters after surgery, such as PCV, TPP, Delta PCV, and Delta TPP; and the presence of reflux were significantly associated with outcome. In addition, we also found that the type of intestinal lesion, resection and anastomosis, duration of surgery, and surgeon experience were also significantly associated with outcome. The multi-variable logistic regression model confirmed that PCV at arrival, TPP after surgery, and BCS had the highest predictive power in a part of the Italian population of horses subjected to colic surgery.

Some of the above reported parameters, however, showed different results to those reported in previous literature, such as age. Furthermore, other parameters have rarely been previously considered, such as BCS, the PCV and TPP values after surgery, Delta PCV and TPP, and, finally, the volume of pre- and intraoperative fluids administered.

In previous studies, the effect of age on outcome has been controversial, with some studies showing that older horses were more likely to die and others reporting that older horses had a similar likelihood of discharge from the hospital than mature horses [4,14]. In this study, the mature horses aged between 10 and 14 years had the best outcome. This can be explained by the fact that young and geriatric animals may have different predisposing conditions, which can also be aggravated by the possible presence of concomitant pathologies.

To the best of our knowledge, BCS has never been related to the short-term outcome. In this study, horses with a lower BCS had a higher risk of a negative outcome. 

This aspect is probably related to the fact that thin horses were debilitated and thus had a reduced ability to recover after colic surgery. Furthermore, also PCV and TPP after surgery, Delta PCV, and Delta TPP had never been evaluated as predictive indices of prognosis. The univariate analysis demonstrated that horses with PCV after surgery < 26% or >43% and a Delta PCV > −8% had a higher risk of non-survival. Similarly, a TPP value after surgery < 4.5 g/dL and a high Delta TPP (>−2.2) resulted in a worse outcome. Moreover, the multi-variable model also showed that a TPP value after surgery < 4.5 g/dL had a higher predictive power because it negatively influenced the short-term outcome. These values could be influenced by several factors, such as type of lesion, intraoperative blood loss, or fluid administration. The multivariate analysis also confirmed the predictive power of PCV upon arrival, TPP after surgery, and BCS as variables associated with a higher risk of a negative outcome.

In the univariate model, the primary lesion also had an effect on the outcome, showing that LCV, small intestinal volvulus, and pedunculated lipoma had a higher probability of a negative outcome. This result was confirmed by the fact that the type of lesion and its location also influenced the outcome, demonstrating that strangulating lesions were associated with a higher likelihood of an adverse outcome. It is important to consider that non-strangulating lesions more frequently occurred in the large intestine [2], and, as reported in previous studies, were associated with a better short-term outcome (>88%) [2,4,5,12,22,23,24,25,26,27,28,29,30,31]. This result could be influenced by the fact that strangulating lesions of the large intestine, such as large colon volvulus, may instead be more commonly associated with pre- or intraoperative death or euthanasia [5,29], resulting in the exclusion of this parameters in the final evaluation. On the other hand, strangulating lesions of the small intestine are associated with an increased risk of a negative outcomes, because they were often recorded in horses operated and judged to be operable, but which had, at the same time, a compromised clinical and haematological status. In addition, strangulating lesions commonly require resection and anastomosis which was reported to be associated with a poorer short-term survival [4]. These findings were confirmed in this study, in which horses that did not have an anastomosis performed had a 2.3 times greater probability of survival than horses with an anastomosis. However, even leaving the strangulated small intestine in place could be dangerous because it could lead to endotoxemia or reperfusion injury and consequently increased risk of post-operative ileus, laminitis, and adhesions [3,4,12,13]. Deciding whether to perform a resection and anastomosis is one of the most important intraoperative factors. Not performing a resection could increase the long-term outcome [3] in cases of viable intestine, but an erroneous judgement may be associated with postoperative complications that could lead to the need for relaparotomy and thus increase the likelihood of a negative outcome [4,32]. Despite the use of a viability index [33] by the surgeons that performed the surgeries in our study to determine whether or not it is necessary to perform an anastomosis, the experience of the surgeon influenced the outcome. This is in accordance with Brown and colleagues, confirming the fact that the mean number of horses operated per year has an effect on the short-term survival rate [26].

Although previous studies have reported that type of anastomosis influenced the outcome [25], this result was not found in our study, which is in accordance with the study by Muller and colleagues [34]. It is important to consider that to compare the effects of the type of anastomosis it would be necessary to evaluate a group of patients presenting with the same type of lesion. The comparison between anastomoses performed on different intestinal tracts could be of limited significance. A strangulating lesion affecting the distal jejunum or ileum could cause a distension of the entire small intestine oral to the lesion and thus the risk of complications and negative outcomes is expected to be greater than in cases in which a lesion is found at the level of the middle or proximal jejunum [7]. Prospective studies comparing different techniques of anastomosis for the same pathology could provide more accurate indications in this regard.

The duration of surgery could also influence the short-term survival rate. As reported in a study by Proudman and colleagues, an increase in duration of surgery was associated with a higher probability of a negative short-term outcome [3]. Moreover, duration of surgery influenced the development of neurological signs during the recovery phase [21,35] or during the postoperative period [36] and predisposed to the development of surgical site infections [37]. However, the duration of surgery is influenced by several factors such as surgeon experience, type of lesion, or decision about whether to perform an intestinal anastomosis or not. These factors that affect the duration of surgery could also influence the outcome or the development of postoperative complications and thus the data concerning the effect of the duration of surgery itself on the outcome may be of marginal significance [4]. For this reason, this parameter was also included in the multi-variable analysis and did not prove to be a good predictive index of prognosis. Despite differences in the population considered in this study compared with that of other studies, the results of this study are generally in line with what is reported in the literature. Furthermore, the short-term outcome for cases included in this multicentre study is comparable to the short-term outcomes reported in previous studies conducted in other Southern European countries such as Spain [3,4,14], but also Northern European countries such as England, the Netherlands, Norway, Finland, and Denmark [5,6,10,38,39,40], as well as the USA [7]. However, it is important to underline the value of these investigations. In fact, the outcomes can be influenced by the intrinsic and extrinsic characteristics of the population considered. Parameters such as weather conditions, training activity, management, and diet, which influence the onset of gastrointestinal problems, can vary considerably from one population to another [14]. Consequently, the results of other studies involving different populations may not necessarily apply to every population. 

In the univariate model, fluid therapy was also considered to influence the outcome. Furthermore, this parameter has rarely been considered in other studies, but can have a significant effect. As already reported in previous studies, the intravenous administration of crystalloids is a life-saving measure, which is fundamental in managing horses with colic [41]. However, the guidelines relating to the volume and rate of infusion are lacking in veterinary medicine [42]. Post-operative treatments are often expensive, and it is essential to reduce the cost of colic surgery through a more selective approach to drug use and a greater emphasis on preventing complications [2,43]. A study by Giusto and colleagues showed that “goal directed” fluid administration is not associated with higher postoperative complications than standard rates of administration, and it has the advantage of being less expensive and reducing hospitalization time [44]. Crystalloid administration influenced the outcome in this study and although there are no studies showing the effect of overhydration on the outcome in horses, the results of our study confirm the need to use a goal directed fluid therapy. Minimal quantities of fluids may be necessary to counteract hypovolemia, hypotension, and hypoperfusion, and ensure normal cardiac output, but excess fluid administration can cause hypervolemia, haemodilution, coagulopathy, pulmonary oedema, and organ dysfunction [36]. Adopting targeted treatments is functional, not only to improve the survival rate, but also to reduce the costs associated with individual treatments or complications. The volume of fluids administered also influences the PCV and TPP values after surgery and consequently also Delta PCV and TPP, as previously explained.

While these variables are also influenced by the horse’s diagnosis and clinical status at arrival and during anaesthesia, a “goal directed fluid therapy” could reduce the risk of overhydration. 

Limitations

The retrospective nature of this analysis limited the possibility to retrieve full data for all of the surgeries performed in the three referral centres.

In this study, not all differences between centres were taken into account, but the overall results were considered. It is important to consider that the weather, the prevalence of breeds, and the activity of horses may differ between Northern and central regions of Italy. Moreover, only effects on short-term survival and not long-term survival were considered. Further studies could be carried out to evaluate the possible differences between the three Italian populations considered. Furthermore, the evaluation of the effects of these predictive indices on long-term survival could lead to different results than those obtained in this study. Further prospective studies should be conducted to confirm the effect of predictive indices considered in this study on short-term survival.

## 5. Conclusions

To the best of our knowledge, this is the first multicentre study in Italy evaluating short-term outcomes of horses subjected to colic surgery. The results of our study suggest that predictive indices previously reported in other studies for different populations are also applicable to the Italian equine population. Body condition score, PCV at arrival, and TPP at the end of surgery may be predictive parameters of short-term survival.

The analysis of these parameters may help surgeons to provide owners with a more accurate prediction of the outcome after colic surgery.

## Figures and Tables

**Figure 1 animals-13-01107-f001:**
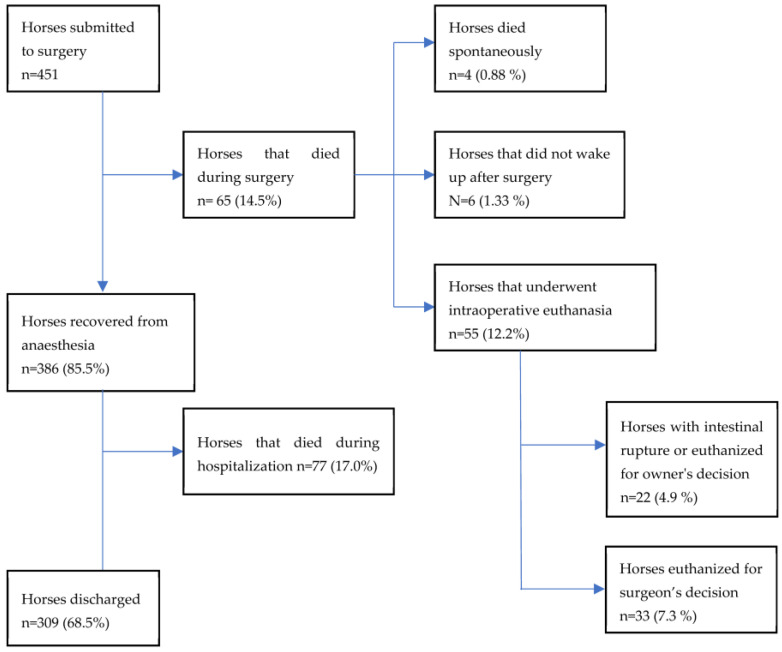
Details of 451 horses submitted to colic surgery and that died intraoperatively or in the postoperative period (percentage calculated based on the number of horses submitted to surgery).

**Figure 2 animals-13-01107-f002:**
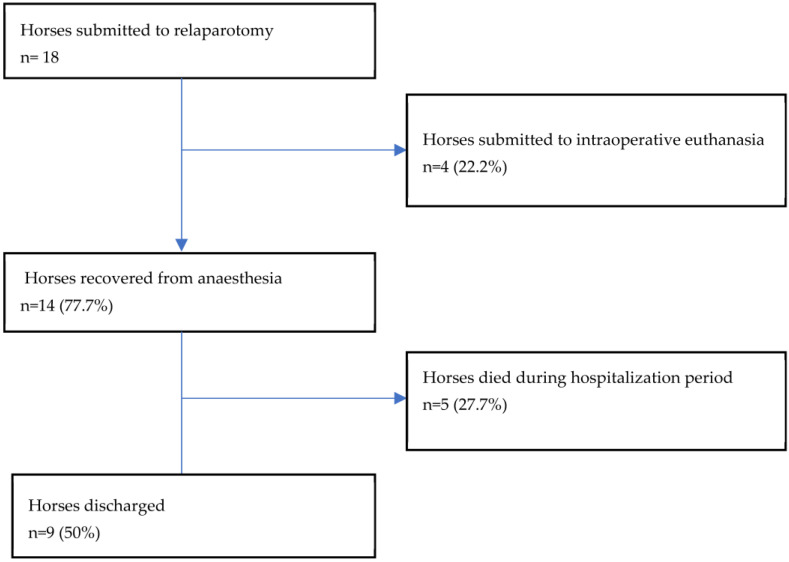
Data on horses submitted to relaparotomy in the same hospitalization period.

**Table 1 animals-13-01107-t001:** Clinical features upon arrival of horses submitted to colic surgery for strangulating intestinal lesions.

Variable	Median	Range	No. Cases with Complete Records
PCV (%)	40	25–72	176
TPP (g/dL)	6.8	3.4–10.5	174
Blood lactate (mmol/L)	3.5	0.9–16.9	128
Reflux (L)	5	0.5–36	41

**Table 2 animals-13-01107-t002:** Clinical features upon arrival of horses submitted to colic surgery for non-strangulating intestinal lesions.

Variable	Median	Range	No. Cases with Complete Records
PCV (%)	39	25–70	259
TPP (g/dL)	7	3–10.9	258
Blood lactate (mmol/L)	2.9	0.7–16.8	184
Reflux (L)	4	0.5–25	57

**Table 3 animals-13-01107-t003:** Duration of symptoms (time (h) between horses (n = 401) last seen healthy and upon arrival at referral hospital.

Hours	No.	%
<4	36	8.97
4–8	87	19.29
8–12	78	17.29
12–16	56	12.41
16–20	28	6.2
20–24	25	5.54
>24	91	22.69

**Table 4 animals-13-01107-t004:** List and % of viscera involved in 451 horses that underwent colic surgery.

Intestine	No.	%
Large colon	215	47.7
Jejunum	91	20.2
Ileum	44	9.7
Jejunum–Ileum	40	8.9
Small colon	24	5.3
Caecum	14	3.1
Duodenum–Jejunum	11	2.4
Mesentery	4	0.9
Stomach–Spleen	3	0.7
Ileum–Caecum	2	0.4
Duodenum	2	0.4
Liver	1	0.2

**Table 5 animals-13-01107-t005:** List and incidence (%) of diagnoses in 451 horses that underwent colic surgery. “Small intestine obstruction” refers to obstructive disorders of the small intestine for which no further details were provided.

Disease	No.	%
Right dorsal displacement (RDD)	71	15.7
Large colon volvulus (LCV)	64	14.1
Nephrosplenic entrapment	42	9.3
Large colon impaction	31	6.9
Epiploic foramen entrapment (EFE)	30	6.7
Pedunculated lipoma	26	5.8
Ileal impaction	22	4.9
Inguinal hernia	21	4.7
Small colon focal impaction	21	4.7
Small intestinal volvulus	18	4.0
Small intestine obstruction	14	3.1
Adhesions	13	2.9
Caecal impaction	9	2.0
Duodenitis/proximal jejunitis (DPJ)	8	1.8
Jejunal impaction	7	1.6
Ileal hypertrophy	6	1.3
Mesenteric rent	5	1.1
Jejuno-jejunal intussusception	5	1.1
Caecal intussusception	3	0.7
Extramural mass	3	0.7
Gastrosplenic entrapment	3	0.7
Omental rent	3	0.7
Extraperitoneal haematoma	2	0.4
Abscess	2	0.4
Diaphragmatic hernia	2	0.4
Mesodiverticular band	2	0.4
Nephrosplenic entrapment of small intestine	2	0.4
Abdominal hernia	1	0.2
Focal eosinophilic enteritis	1	0.2
Hepatic abscess	1	0.2
Ileo-caecal intussusception	1	0.2
Jejuno-ileo-caecal intussusception	1	0.2
Caecal hypertrophy	1	0.2
Mesoduodenal entrapment	1	0.2
Gastric impaction	1	0.2
IV grade rectal prolaps	1	0.2
Diffused Lymphoma	1	0.2
Mesenteric abscess	1	0.2
Mesenteric ischemia	1	0.2
Small intestinal leyomioma	1	0.2
Typhlitis	1	0.2

**Table 6 animals-13-01107-t006:** List and % of types of anastomoses reported in 108 horses that underwent colic surgery.

Anastomosis	No.	%
Jejuno-jejunal (JJ)	38	35.1
Jejuno-caecal bypass (JCE BYPASS)	22	20.4
Jejuno-caecal (JCE)	21	19.4
Jejuno-ileal (JI)	8	7.4
Hybrid jejuno-ileo caecal (HJICE)	7	6.3
Colo-colic (COCO)	6	5.6
End-to-end of small colon (SCO-SCO)	4	3.7
Jejuno-ileo-colic (JICO)	2	1.9

**Table 7 animals-13-01107-t007:** Postoperative clinical features of horses recovered from anaesthesia after colic surgery. Delta PCV and Delta TPP are the difference between PCV and TPP, respectively, at arrival and soon after recovery from surgery.

Variable	Median	Range	No. Cases Recorded
PCV (%)	34	22–64	206
TPP (g/dL)	5.4	3.2–8.1	205
Delta PCV	−6	−45–36	207
Delta TPP	−1.4	−7.8–1.7	202

**Table 8 animals-13-01107-t008:** List and % of postoperative complications reported in 386 horses recovered after colic surgery.

Complication	No.	%
Pyrexia	107	27.7
Surgical site infection (any discharge from wound)	97	25.1
Postoperative colic	91	23.5
Clinical signs of piroplasmosis (regenerative anemia, fever, with or without positive PCR or confirmed presence of piroplasms in a blood smear)	88	22.8
Postoperative reflux	79	20.4
Diarrhoea	39	10.1
Thrombophlebitis	30	7.7
Incisional hernia	24	5.7
Other complications (reported but not detailed)	23	5.9
SIRS	17	4.4
Laminitis	12	3.1
Hyperlipemia	8	2
Wound dehiscence	7	1.8
Myopathy	7	1.8
Hemoperitoneum	1	0.2

**Table 9 animals-13-01107-t009:** Outcome of the 5 most commonly reported primary lesions affecting the large (^a^, *p* = 0.0018; ^b^, *p* = 0.0153; ^c^, *p* = 0.0055) and small intestine (^d^, *p* = 0.0264; ^e^, *p* = 0.045).

Pathology	No. Cases Underwent Colic Surgery	No. Cases Underwent Intraoperative Death/Euthanasia	No. Cases Hospitalized	No. Cases Discharged	% Short-Term Survival
Nephrosplenic entrapment	42	0	42	37	88% ^a^
Large colon impaction	31	2	29	26	83.8% ^b^
Right dorsal displacement (RDD)	71	3	66	57	80.2% ^c^
Small colon focal impaction	21	4	17	15	71.4%
Large colon volvulus (LCV)	64	11	51	37	57.8% ^a,b,c^
Inguinal hernia	21	0	21	18	85.7% ^d,e^
Ileal impaction	22	0	22	17	77.2%
Epiploic foramen entrapment (EFE)	30	5	25	19	63.3%
Small intestinal volvulus	18	6	11	10	55.5% ^d^
Pedunculated lipoma	26	6	19	14	53.8% ^e^

**Table 10 animals-13-01107-t010:** Details on surgeon’s experience and the effect on short-term outcome.

Surgeon	No. of Colic Surgeries per Surgeon per Year (Mean)	Colic Surgeries per Surgeon per Year (%)	Intraop. Euthanasia (%)	Relap. (%)	Postop. Euthanasia (%)	Discharged/ Hospitalized (%)	Discharged/ Total (%)
1	13.1	9.5	16.3	2.8	25.0	72.2	60.5
3	9.6	5.3	29.2	0.0	29.4	70.6	50.0
2	4.8	3.5	12.5	14.3	21.4	64.3	56.3
4	39.6	33.3	13.2	6.1	19.1	74.8	64.9
5	18	5.7	0.0	0.0	11.5	88.5	88.5
6	69	42.6	13.0	4.8	12.5	82.7	72.0
						r = 0.83	r = 0.77

**Table 11 animals-13-01107-t011:** Results of the multivariate logistic regression model used to evaluate the association between the independent variables and negative outcome in 386 horses hospitalized after colic surgery.

Variable	Cut-Off/Negative Outcome	Odds Ratio	95% CI	*p*-Value
TPP after surgery (g/dL)	<4.5	2.83	1.15–6.92	0.015
BCS	>7	0.76	0.63–0.93	0.043
PCV on arrival (%)	<50	0.29	0.15–0.57	0.054

## Data Availability

Not applicable.

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
