# Peer review of "Short-Term Survival and Postoperative Complications Rates in Horses Undergoing Colic Surgery: A Multicentre Study"

_animals, 2023, doi:10.3390/ani13061107_

Round 1
Reviewer 1 Report
Overall, this paper confirms what has previously been published in the literature on this subject. If the authors feel that further publication in the field is justified, further explanation should be provided as to why Italian horses would be any different than other populations of horses. Additionally, the conclusion regarding administration of crystalloids is very misleading to the reader, and likely seems to be correlative rather than causative.
General comment: Please review grammar and punctuation, per the journal’s guidelines as there are numerous errors throughout the current version of this manuscript.
Simple Summary
Line 13: Not all colic cases require surgical treatment, more intensive medical management may also be sufficient.
Lines 23-24: These comments about fluids are not helpful for the reader. Additionally, if the animal is well-hydrated then fluids may not be necessary. Additionally, IV vs enteral fluids is not specified, and should be balanced against the additional risks for administration.
Abstract
Lines 38-39: The comments about fluid administration are somewhat misleading, as often shock boluses are given in the amount of 40-60 ml/kg. Additionally, is selection bias a possibility, i.e., are horses with more severe lesions or in more critical condition being given larger fluid boluses? The causation/correlation relationship should be carefully addressed.
Line 42: Why are these outcomes only relevant to the Italian horse population? Why were they expected to be any different in the first place?
Introduction
Lines 48-49: Please clarify this statement. Are you referring the anesthetic-related deaths or underlying etiology for colic-related deaths?
Lines 56-58: There are numerous other publications looking at short-term outcomes in different types of colic.
Line 59: Please clarify why the Italian population of horses are believed to be different than other parts of the world. It is difficult to understand the impact of this paper without that information. This comment also applies to lines 68-69.
Materials & Methods
Line 78: Were there inclusion or exclusion criteria? Did records need to be complete or were incomplete records acceptable? Please expand the details here.
Line 97: If surgeon experience if of interest, this should be presented as captured information for the retrospective study.
Results
Line 114: Although general details can be helpful, it would likely be more helpful for the reader if horses and admission data were divided based on final diagnosis, since strangulating lesions will have markedly different clinical and hematologic admission data compared to non-strangulating lesion cases.
Line 131: Did all horses die related to colic? Were there catastrophic injuries or other medical complications? Please specify.
Lines 130-142: These data are likely better described in a table. These data may be integrated into Table 2, or the table modified to a figure/flow chart.
Lines 145-147: Data provided in a table should not be repeated in the text.
Lines 152-154: See previous comment.
Lines 161-164: Why do the percentages in the text not match the table?
General comment: The tables in the results are basic lists and could be reworked and integrated to provide more valuable information for the reader. Please also provide table legends.
Table 5: Why are some surgeons bolded? Presumably this has to do with their experience, but no table legends (for this table and others) are provided.
Lines 184-185: What is this time period?
Lines 215-216: Did these older horses have a higher proportion of a specific type of lesion?
Line 238: Please provide more clarification on this statement—was this across all hospitalization? Immediately pre-op? Is there a correlation to PCV (assumed)? Were crystalloids used in conjunction with hypertonic saline or other colloids?
Discussion
General comment: Overall, the Discussion feels to be strongly repeating what has been published elsewhere. It is difficult to see the “value added” to the reader from the current discussion.
Lines 322-323: The rate and volume of infusion is described in the veterinary literature and the human literature. These comments should be tempered/revised.
Conclusions
General comments: The authors should address why the Italian population of horses is expected to be different and the impact this study has on the body of clinical literature as it relates to equine colic.
Author Response
REVIEWER 1:
Simple Summary
R1.1 Line 13: Not all colic cases require surgical treatment, more intensive medical management may also be sufficient.
The reviewer is right. This is included in “medical treatment”
R1.2 Lines 23-24: These comments about fluids are not helpful for the reader. Additionally, if the animal is well-hydrated then fluids may not be necessary. Additionally, IV vs enteral fluids is not specified, and should be balanced against the additional risks for administration.
R1.2 the whole article has been reviewed according this comment and other comments. This issue has been completely rewritten. Abstract has been revised accordignly
Abstract
R1.3 Lines 38-39: The comments about fluid administration are somewhat misleading, as often shock boluses are given in the amount of 40-60 ml/kg. Additionally, is selection bias a possibility, i.e., are horses with more severe lesions or in more critical condition being given larger fluid boluses? The causation/correlation relationship should be carefully addressed.
R1.3 the reviewer is right. The abstract has been revised
R1.4 Line 42: Why are these outcomes only relevant to the Italian horse population? Why were they expected to be any different in the first place?
R1.4
Changed, see text
Introduction
R1.5 Lines 48-49: Please clarify this statement. Are you referring the anesthetic-related deaths or underlying etiology for colic-related deaths?
R1.5: changed, see text
R1.6 Lines 56-58: There are numerous other publications looking at short-term outcomes in different types of colic.
Changed, see text
R1.7 Line 59: Please clarify why the Italian population of horses are believed to be different than other parts of the world. It is difficult to understand the impact of this paper without that information. This comment also applies to lines 68-69.
R1.7 changed, see text
Materials & Methods
R1.8 Line 78: Were there inclusion or exclusion criteria? Did records need to be complete or were incomplete records acceptable? Please expand the details here.
Inserted, see text
R1.9 Line 97: If surgeon experience if of interest, this should be presented as captured information for the retrospective study.
Inserted, see text
Results
R1.10 Line 114: Although general details can be helpful, it would likely be more helpful for the reader if horses and admission data were divided based on final diagnosis, since strangulating lesions will have markedly different clinical and hematologic admission data compared to non-strangulating lesion cases.
Inserted, see text
R1.11 Line 131: Did all horses die related to colic? Were there catastrophic injuries or other medical complications? Please specify.
This is specified at the end of paragraph
R1.12 Lines 130-142: These data are likely better described in a table. These data may be integrated into Table 2, or the table modified to a figure/flow chart.
A flow chart has been inserted
R1.13 Lines 145-147: Data provided in a table should not be repeated in the text.
Changed, see text
R1.14 Lines 152-154: See previous comment.
Changed, see text
R1.15 Lines 161-164: Why do the percentages in the text not match the table?
Sorry, data have been re-checked
R1.16 General comment: The tables in the results are basic lists and could be reworked and integrated to provide more valuable information for the reader. Please also provide table legends.
Changed, see table
Table 5: Why are some surgeons bolded? Presumably this has to do with their experience, but no table legends (for this table and others) are provided.
Changed, see table
R1.17 Lines 184-185: What is this time period?
The reviewer is right. We inserted this detail (within the same hospitalization period)
R1.18 Lines 215-216: Did these older horses have a higher proportion of a specific type of lesion?
Inserted in text
R1.19 Line 238: Please provide more clarification on this statement—was this across all hospitalization? Immediately pre-op? Is there a correlation to PCV (assumed)? Were crystalloids used in conjunction with hypertonic saline or other colloids?
Inserted, see text
Discussion
R1.20 General comment: Overall, the Discussion feels to be strongly repeating what has been published elsewhere. It is difficult to see the “value added” to the reader from the current discussion.
R1.20 the whole introduction and discussion manuscript have been revised and multivariate analysis has been performed to add value to the manuscript
R1.21 Lines 322-323: The rate and volume of infusion is described in the veterinary literature and the human literature. These comments should be tempered/revised.
Changed, see text
Conclusions
R1.22 General comments: The authors should address why the Italian population of horses is expected to be different and the impact this study has on the body of clinical literature as it relates to equine colic.
R1.22 changed, see text

Reviewer 2 Report
some minor changes, see the following notes.
line 24: delete "."
page 6 first line substitute DPJ with "Duodenitis-proximal jeunitis (DPJ)"
page 6 line 16th: substitute POI with "Post operative ileus (POI)"
page 9, last line: substitute SIRS with "Systemic inflammatory response syndrome (SIRS)"
Author Response
REVIEWER 2:
R2.1 Line 11: still has
Changed, see text
R2.2 Line 19: 80% of the horses surviving surgery
Changed, see text
R2.3 Line 20: in line with previously published data
Changed, see text
R2.4 Line 21: intra and postoperatively
Changed, see text
R2.5 Line 21: the outcome. This
R2.6 Line 22: on the Italian equine population
Changed, see text
R2.7 Line 23 and 24: Pease check punctuation, spaces and capital characters
Changed, see text
R2.8 Line 33 and 34: Please adjust this sentence. Meaning is currently unclear
Changed,see text
R2.9 Line 36: 80% of the horses surviving surgery
Changed, see text
R2.10 Line 38: in line with previously published data
Changed, see text
R2.11 Line 42: on the Italian equine population
Changed, see text
R2.12 Line 48: still has
Changed, see text
R2.13 Line 53: have already been published
Changed, see text
R2.14 Line 54: short-term. Please be consistent
Changed, see text
R2.15 Line 57: please rephrase; this assumption is not really true since some of your references are from 2020 and 2022
Changed, see text
R2.16 Line 58 and 59: you could say that no such multicentric studies have been conducted in Italy
Changed, see text
R2.17 Line 62: has its own intrinsic and…
Changed, see text
R2.18 Line 66: Studies on predictive
Changed, see text
R2.19 Line 88: if horses survived to discharge or not.
Changed, see text
R2.20 Line 91: statistical software (JMP Pro 16, JMP Statistical Discovery LLC).
Changed, see text
R2.21 Line 93: of all horses underwent colic surgery
Changed, see text
R2.22 Line 94 and 95: Please adjust. Maybe say the differences between horses surviving and not surviving to discharge were…
Changed, see text
R2.23 Line 100 and 101: were used in parameters that were significant in the univariate analysis to evaluate discrimination…
Changed, see text
R2.24 Line 103: in the previous model OR in previous models
Changed, see text
R2.25 Lines 111-113: Maybe put only one number after the dot in your percentages to be consistent
Changed, see text
R2.26 Line 111: What is a hot blood horse? Thoroughbred?
Hot-blooded horses include thoroughbreds and arabians
R2.27 Line 112: Saddlebred
Changed, see text
R2.28 Line 114: Clinical features upon arrival
Changed, see text
R2.29 Line 116: between brackets
Changed, see text
R2.30 Line 120: Nasogastric reflux
Changed, see text
R2.31 Line 121: Please adjust. I don’t understand the 22.6% if you say that MOST of the horses were seen healthy during the previous 24 hours
The reviewer is right. N of horses was 91 out of 401 (22.7%) for which this data was available
R2.32 Line 122: to the hospital; when the animal was last seen healthy
changed, see text
R2.33 Table 1: It would be more logical to organize the table with increasing hours, not with increasing numbers
Changed, see text
R2.34 Line 130: we are missing 0.1%
Changed see text
R2.35 Line 131: in the recovery box
Changed, see text
R2.36 Line 132 and 133: you should explain what the difference is between owner’s decision and surgeon’s decision and adjust the term “not operable” because we actually DID operate on them
Changed, see text
R2.37 Line 136: of the small intestine; of the large intestine…
Changed, see text
R2.38 Line 138: please explain what are the 2 cases of other viscera
Inserted, see text
R2.39 Line 140: I doubt that endotoxemia was a surgical finding justifying euthanasia. Please rephrase
Changed, see text
R2.40 Line 141: did not recover from anesthesia
Changed, see text
R2.41 Lines 143-144: still missing 0.1%
Changed, see text
R2.42 Line 149, 156 and 166: that underwent colic surgery
Changed, see text
R2.43 Table 2: why is the pelvic flexure separated from the large colon??
Sorry, this was a mistake. Pelvic flexure has been included with large colon
R2.44 Table 3: please develop DPJ; why did you separate small intestine obstruction, ileal impaction and jejunal impaction? Was the abdominal hernia the cause of the colic requiring a laparotomy? What is a duodenal entrapment? Rectal prolapse. Why did you separate mesenteric abscess, abscess and hepatic abscess? Why did you separate small intestinal neoplasia, lymphoma and focal eosinophilic enteritis?
R2.45 Line 161: resection and anastomosis were performed
Changed, see text
R2.46 Table 4: Hybrid; I would say jejuno-jejunal rather that jejuno-jejunostomy to be consistent
Changed, see text
R2.47 Line 170: add “hours”; how can it be 0 for the lowest part of the range?? I takes at least a few minutes to open up an equine abdomen
Inserted and corrected, see text
R2.48 Table 5: in the third column you should maybe say “Number of colic surgeries per year”
Changed, see text
R2.49 Line 180: please define delta PCV and delta TPP
Inserted, see text
R2.50 Line 192: please define pyrexia: 38,6°C?
Inserted, see text
R2.51 Line 193: please rephrase, not clear
Changed, see text
R2.52 Line 194: please define SSI: drainage? Purulent? Sanguineous?
Inserted, see text
R2.53 Line 195: Please define clinical signs of piroplasmosis
Inserted, see text
R2.54 Table 6: your percentage of pyrexia is different between the text and the table; same problem for piroplasmosis, colic, reflux… Why did you separate piroplasmosis and clinical signs of piroplasmosis? What are the other complications?
Changed, see text
R2.55 Line 205: that underwent colic surgery
Changed, see text
R2.56 Line 206: 386 horses that survived surgery
Changed, see text
R2.57 Line 207: that underwent euthanasia
Changed, see text
R2.58 Lines 208-210: please rephrase
Changed, see text
R2.59 Lines 212-213 should belong to the discussion
Changed, see text
R2.60 Line 215: Horses aged >14 years were 2.3 times more likely…
Changed, see text
R2.61 Line 218: please rephase “The cut-off to define the risk BCS value is 7”
Changed, see text
R2.62 Line 222: had a poorer prognosis
Changed, see text
R2.63 Line 226: associated with a higher
Changed, see text
R2.64 Line 231: resection and anastomosis
Changed, see text
R2.65 Line 232: higher than in horses that underwent intestinal
Changed, see text
R2.66 Line 233: The type of anastomosis did not have a significant effect on the outcome.
Changed, see text
R2.67 Line 235: numbers of horses operated per year? Please rephrase
Changed, see text
R2.68 Line 238: associated with a poorer prognosis
Changed, see text
R2.69 Line 239: please remove “in conclusion”
Changed, see text
R2.70 Line 242: how long do you need for your delta PCV? How many hours after surgery?
Described above, see text
R2.71 Line 244: is it really a loss < or > to 2.2g/dL?
The reviewer is right. Changed, see text
R2.72 Line 260: horses that underwent OR horses undergoing
Changed, see text
R2.73 Line 271: Freeman 2018 should have a reference number
Changed, see text
R2.74 Lines 271-272: What is the difference between a favourable prognosis and a positive short-term outcome. Please clarify
Changed, see text
R2.75 Lines 274-275: “which is why they may have less impact on the short-term survival rate” this sentence is unclear. Please clarify
Removed, see text
R2.76 Line 279 and 280: of the small intestine
Changed, see text
R2.77 Line 282: horses that did not have an anastomosis performed
Changed, see text
R2.78 Line 284: because it could cause…Well, in this sentence it must be clarified that a necrotic piece of bowel has to be resected in all cases and that in other cases, if the strangulated piece of bowel is deemed viable it can be left in place. Please refer to Freeman et al., EVJ 2014. In this paper different grades are defined to assess viability of the strangulated small intestine. Same thing for line 289-293
This paragraph has been removed
R2.79 Line 294: The type of anastomosis was not found to affect the outcome
Changed, see text
R2.80 Lines 304-305: “Prospective studies comparing different types of anastomoses for the same pathology could give more indications in this regard”. As you mention in line 298, the type of anastomosis depends on the actual pathology and most of the time you don’t really have the choice about the anastomosis that has to be done. Some papers have compared the jejunoileal anastomosis and the jejunocecal anastomosis, maybe you could discuss/add this. Please adjust this part.
Changed, see text.
R2.81 Line 309: could influence the development; during recovery
Changed, see text
R2.82 Line 311-313: your statistical analysis should tell you if surgical time is a confounding factor or really a risk factors for complications to happen
Changed, see text
R2.83 Lines 320-338: As explained previously, this part must be balanced or adjusted. Sick horses, such as those with a LCV or a small intestinal resection and anastomosis, are likely in need of more fluids than horses with a simple large colon displacement. These sick horses are also less likely to survive from their compromised status. To me it is an overstatement to say that “overhydration” is a negative factor from survival.
R2.83. this part has been adjusted
R2.84 A multivariate analysis could greatly improve the conclusions of your study.
It has been included in the manuscript

Reviewer 3 Report
Overall, the conclusion that excessive fluid therapy worsens the prognosis seems unsuitable. It is much more likely that critically sick horses would require more fluids, a longer hospitalization period and would be more likely to die.
Line 11: still has
Line 19: 80% of the horses surviving surgery
Line 20: in line with previously published data
Line 21: intra and postoperatively
Line 21: the outcome. This
Line 22: on the Italian equine population
Line 23 and 24: Pease check punctuation, spaces and capital characters
Line 33 and 34: Please adjust this sentence. Meaning is currently unclear
Line 36: 80% of the horses surviving surgery
Line 38: in line with previously published data
Line 42: on the Italian equine population
Line 48: still has
Line 53: have already been published
Line 54: short-term. Please be consistent
Line 57: please rephrase; this assumption is not really true since some of your references are from 2020 and 2022
Line 58 and 59: you could say that no such multicentric studies have been conducted in Italy
Line 62: has its own intrinsic and…
Line 66: Studies on predictive
Line 88: if horses survived to discharge or not.
Line 91: statistical software (JMP Pro 16, JMP Statistical Discovery LLC).
Line 93: of all horses underwent colic surgery
Line 94 and 95: Please adjust. Maybe say the differences between horses surviving and not surviving to discharge were…
Line 100 and 101: were used in parameters that were significant in the univariate analysis to evaluate discrimination…
Line 103: in the previous model OR in previous models
Lines 111-113: Maybe put only one number after the dot in your percentages to be consistent
Line 111: What is a hot blood horse? Thoroughbred?
Line 112: Saddlebred
Line 114: Clinical features upon arrival
Line 116: between brackets
Line 120: Nasogastric reflux
Line 121: Please adjust. I don’t understand the 22.6% if you say that MOST of the horses were seen healthy during the previous 24 hours
Line 122: to the hospital; when the animal was last seen healthy
Table 1: It would be more logical to organize the table with increasing hours, not with increasing numbers
Line 130: we are missing 0.1%
Line 131: in the recovery box
Line 132 and 133: you should explain what the difference is between owner’s decision and surgeon’s decision and adjust the term “not operable” because we actually DID operate on them
Line 136: of the small intestine; of the large intestine…
Line 138: please explain what are the 2 cases of other viscera
Line 140: I doubt that endotoxemia was a surgical finding justifying euthanasia. Please rephrase
Line 141: did not recover from anesthesia
Lines 143-144: still missing 0.1%
Line 149, 156 and 166: that underwent colic surgery
Table 2: why is the pelvic flexure separated from the large colon??
Table 3: please develop DPJ; why did you separate small intestine obstruction, ileal impaction and jejunal impaction? Was the abdominal hernia the cause of the colic requiring a laparotomy? What is a duodenal entrapment? Rectal prolapse. Why did you separate mesenteric abscess, abscess and hepatic abscess? Why did you separate small intestinal neoplasia, lymphoma and focal eosinophilic enteritis?
Line 161: resection and anastomosis were performed
Table 4: Hybrid; I would say jejuno-jejunal rather that jejuno-jejunostomy to be consistent
Line 170: add “hours”; how can it be 0 for the lowest part of the range?? I takes at least a few minutes to open up an equine abdomen
Table 5: in the third column you should maybe say “Number of colic surgeries per year”
Line 180: please define delta PCV and delta TPP
Line 192: please define pyrexia: 38,6°C?
Line 193: please rephrase, not clear
Line 194: please define SSI: drainage? Purulent? Sanguineous?
Line 195: Please define clinical signs of piroplasmosis
Table 6: your percentage of pyrexia is different between the text and the table; same problem for piroplasmosis, colic, reflux… Why did you separate piroplasmosis and clinical signs of piroplasmosis? What are the other complications?
Line 205: that underwent colic surgery
Line 206: 386 horses that survived surgery
Line 207: that underwent euthanasia
Lines 208-210: please rephrase
Lines 212-213 should belong to the discussion
Line 215: Horses aged >14 years were 2.3 times more likely…
Line 218: please rephase “The cut-off to define the risk BCS value is 7”
Line 222: had a poorer prognosis
Line 226: associated with a higher
Line 231: resection and anastomosis
Line 232: higher than in horses that underwent intestinal
Line 233: The type of anastomosis did not have a significant effect on the outcome.
Line 235: numbers of horses operated per year? Please rephrase
Line 238: associated with a poorer prognosis
Line 239: please remove “in conclusion”
Line 242: how long do you need for your delta PCV? How many hours after surgery?
Line 244: is it really a loss < or > to 2.2g/dL?
Line 260: horses that underwent OR horses undergoing
Line 271: Freeman 2018 should have a reference number
Lines 271-272: What is the difference between a favourable prognosis and a positive short-term outcome. Please clarify
Lines 274-275: “which is why they may have less impact on the short-term survival rate” this sentence is unclear. Please clarify
Line 279 and 280: of the small intestine
Line 282: horses that did not have an anastomosis performed
Line 284: because it could cause…
Well, in this sentence it must be clarified that a necrotic piece of bowel has to be resected in all cases and that in other cases, if the strangulated piece of bowel is deemed viable it can be left in place. Please refer to Freeman et al., EVJ 2014. In this paper different grades are defined to assess viability of the strangulated small intestine. Same thing for line 289-293
Line 294: The type of anastomosis was not found to affect the outcome
Lines 304-305: “Prospective studies comparing different types of anastomoses for the same pathology could give more indications in this regard”. As you mention in line 298, the type of anastomosis depends on the actual pathology and most of the time you don’t really have the choice about the anastomosis that has to be done. Some papers have compared the jejunoileal anastomosis and the jejunocecal anastomosis, maybe you could discuss/add this. Please adjust this part.
Line 309: could influence the development; during recovery
Line 311-313: your statistical analysis should tell you if surgical time is a confounding factor or really a risk factors for complications to happen
Lines 320-338: As explained previously, this part must be balanced or adjusted. Sick horses, such as those with a LCV or a small intestinal resection and anastomosis, are likely in need of more fluids than horses with a simple large colon displacement. These sick horses are also less likely to survive from their compromised status. To me it is an overstatement to say that “overhydration” is a negative factor from survival.
A multivariate analysis could greatly improve the conclusions of your study.
Author Response
REVIEWER 3:
R3.1 Line 11: still has
Changed, see text
R3.2 Line 19: 80% of the horses surviving surgery
Changed, see text
R3.3 Line 20: in line with previously published data
Changed, see text
R3.4 Line 21: intra and postoperatively
Changed, see text
R3.5 Line 21: the outcome. This
R3.6 Line 22: on the Italian equine population
Changed, see text
R3.7 Line 23 and 24: Pease check punctuation, spaces and capital characters
Changed, see text
R3.8 Line 33 and 34: Please adjust this sentence. Meaning is currently unclear
Changed,see text
R3.9 Line 36: 80% of the horses surviving surgery
Changed, see text
R3.10 Line 38: in line with previously published data
Changed, see text
R3.11 Line 42: on the Italian equine population
Changed, see text
R3.12 Line 48: still has
Changed, see text
R3.13 Line 53: have already been published
Changed, see text
R3.14 Line 54: short-term. Please be consistent
Changed, see text
R3.15 Line 57: please rephrase; this assumption is not really true since some of your references are from 2020 and 2022
Changed, see text
R3.16 Line 58 and 59: you could say that no such multicentric studies have been conducted in Italy
Changed, see text
R3.17 Line 62: has its own intrinsic and…
Changed, see text
R3.18 Line 66: Studies on predictive
Changed, see text
R3.19 Line 88: if horses survived to discharge or not.
Changed, see text
R3.20 Line 91: statistical software (JMP Pro 16, JMP Statistical Discovery LLC).
Changed, see text
R3.21 Line 93: of all horses underwent colic surgery
Changed, see text
R3.22 Line 94 and 95: Please adjust. Maybe say the differences between horses surviving and not surviving to discharge were…
Changed, see text
R3.23 Line 100 and 101: were used in parameters that were significant in the univariate analysis to evaluate discrimination…
Changed, see text
R3.24 Line 103: in the previous model OR in previous models
Changed, see text
R3.25 Lines 111-113: Maybe put only one number after the dot in your percentages to be consistent
Changed, see text
R3.26 Line 111: What is a hot blood horse? Thoroughbred?
Hot-blooded horses include thoroughbreds and arabians
R3.27 Line 112: Saddlebred
Changed, see text
R3.28 Line 114: Clinical features upon arrival
Changed, see text
R3.29 Line 116: between brackets
Changed, see text
R3.30 Line 120: Nasogastric reflux
Changed, see text
R3.31 Line 121: Please adjust. I don’t understand the 22.6% if you say that MOST of the horses were seen healthy during the previous 24 hours
The reviewer is right. N of horses was 91 out of 401 (22.7%) for which this data was available
R3.32 Line 122: to the hospital; when the animal was last seen healthy
changed, see text
R3.33 Table 1: It would be more logical to organize the table with increasing hours, not with increasing numbers
Changed, see text
R3.34 Line 130: we are missing 0.1%
Changed see text
R3.35 Line 131: in the recovery box
Changed, see text
R3.36 Line 132 and 133: you should explain what the difference is between owner’s decision and surgeon’s decision and adjust the term “not operable” because we actually DID operate on them
Changed, see text
R3.37 Line 136: of the small intestine; of the large intestine…
Changed, see text
R3.38 Line 138: please explain what are the 2 cases of other viscera
Inserted, see text
R3.39 Line 140: I doubt that endotoxemia was a surgical finding justifying euthanasia. Please rephrase
Changed, see text
R3.40 Line 141: did not recover from anesthesia
Changed, see text
R3.41 Lines 143-144: still missing 0.1%
Changed, see text
R3.42 Line 149, 156 and 166: that underwent colic surgery
Changed, see text
R3.43 Table 2: why is the pelvic flexure separated from the large colon??
Sorry, this was a mistake. Pelvic flexure has been included with large colon
R3.44 Table 3: please develop DPJ; why did you separate small intestine obstruction, ileal impaction and jejunal impaction? Was the abdominal hernia the cause of the colic requiring a laparotomy? What is a duodenal entrapment? Rectal prolapse. Why did you separate mesenteric abscess, abscess and hepatic abscess? Why did you separate small intestinal neoplasia, lymphoma and focal eosinophilic enteritis?
R3.45 Line 161: resection and anastomosis were performed
Changed, see text
R3.46 Table 4: Hybrid; I would say jejuno-jejunal rather that jejuno-jejunostomy to be consistent
Changed, see text
R3.47 Line 170: add “hours”; how can it be 0 for the lowest part of the range?? I takes at least a few minutes to open up an equine abdomen
Inserted and corrected, see text
R3.48 Table 5: in the third column you should maybe say “Number of colic surgeries per year”
Changed, see text
R3.49 Line 180: please define delta PCV and delta TPP
Inserted, see text
R3.50 Line 192: please define pyrexia: 38,6°C?
Inserted, see text
R3.51 Line 193: please rephrase, not clear
Changed, see text
R3.52 Line 194: please define SSI: drainage? Purulent? Sanguineous?
Inserted, see text
R3.53 Line 195: Please define clinical signs of piroplasmosis
Inserted, see text
R3.54 Table 6: your percentage of pyrexia is different between the text and the table; same problem for piroplasmosis, colic, reflux… Why did you separate piroplasmosis and clinical signs of piroplasmosis? What are the other complications?
Changed, see text
R3.55 Line 205: that underwent colic surgery
Changed, see text
R3.56 Line 206: 386 horses that survived surgery
Changed, see text
R3.57 Line 207: that underwent euthanasia
Changed, see text
R3.58 Lines 208-210: please rephrase
Changed, see text
R3.59 Lines 212-213 should belong to the discussion
Changed, see text
R3.60 Line 215: Horses aged >14 years were 2.3 times more likely…
Changed, see text
R3.61 Line 218: please rephase “The cut-off to define the risk BCS value is 7”
Changed, see text
R3.62 Line 222: had a poorer prognosis
Changed, see text
R3.63 Line 226: associated with a higher
Changed, see text
R3.64 Line 231: resection and anastomosis
Changed, see text
R3.65 Line 232: higher than in horses that underwent intestinal
Changed, see text
R3.66 Line 233: The type of anastomosis did not have a significant effect on the outcome.
Changed, see text
R3.67 Line 235: numbers of horses operated per year? Please rephrase
Changed, see text
R3.68 Line 238: associated with a poorer prognosis
Changed, see text
R3.69 Line 239: please remove “in conclusion”
Changed, see text
R3.70 Line 242: how long do you need for your delta PCV? How many hours after surgery?
Described above, see text
R3.71 Line 244: is it really a loss < or > to 2.2g/dL?
The reviewer is right. Changed, see text
R3.72 Line 260: horses that underwent OR horses undergoing
Changed, see text
R3.73 Line 271: Freeman 2018 should have a reference number
Changed, see text
R3.74 Lines 271-272: What is the difference between a favourable prognosis and a positive short-term outcome. Please clarify
Changed, see text
R3.75 Lines 274-275: “which is why they may have less impact on the short-term survival rate” this sentence is unclear. Please clarify
Removed, see text
R3.76 Line 279 and 280: of the small intestine
Changed, see text
R3.77 Line 282: horses that did not have an anastomosis performed
Changed, see text
R3.78 Line 284: because it could cause…Well, in this sentence it must be clarified that a necrotic piece of bowel has to be resected in all cases and that in other cases, if the strangulated piece of bowel is deemed viable it can be left in place. Please refer to Freeman et al., EVJ 2014. In this paper different grades are defined to assess viability of the strangulated small intestine. Same thing for line 289-293
This paragraph has been removed
R3.79 Line 294: The type of anastomosis was not found to affect the outcome
Changed, see text
R3.80 Lines 304-305: “Prospective studies comparing different types of anastomoses for the same pathology could give more indications in this regard”. As you mention in line 298, the type of anastomosis depends on the actual pathology and most of the time you don’t really have the choice about the anastomosis that has to be done. Some papers have compared the jejunoileal anastomosis and the jejunocecal anastomosis, maybe you could discuss/add this. Please adjust this part.
Changed, see text.
R3.81 Line 309: could influence the development; during recovery
Changed, see text
R3.82 Line 311-313: your statistical analysis should tell you if surgical time is a confounding factor or really a risk factors for complications to happen
Changed, see text
R3.83 Lines 320-338: As explained previously, this part must be balanced or adjusted. Sick horses, such as those with a LCV or a small intestinal resection and anastomosis, are likely in need of more fluids than horses with a simple large colon displacement. These sick horses are also less likely to survive from their compromised status. To me it is an overstatement to say that “overhydration” is a negative factor from survival.
R3.83. this part has been adjusted
R3.84 A multivariate analysis could greatly improve the conclusions of your study.
It has been included in the manuscript

Reviewer 4 Report
Table 1 : would make more sense to call it : duration of symptoms.
Please adjust the time list in the table in a more logical manner.
Please be consistent throughout the manuscript with the number expressed as numeric and not as spelled in words.
Section on reasons for euthanasia starting in line 133 can be deleted.
I would suggest the result section to consist of less text description and more tables.
In your statistics I am missing torsion of the large colon- but in the section line 159 you refer to strangulating lesions. If you had both types of volvolus of the large colon in your case load, you need to separate them into non-strangulating volvolus of the large colon and large colon torsion, sometimes also referred to as volvolus and indicate the percentages of these also separatly in the tables.
Please always spell out abbreviations in the table legend.
Table 5 is incomprehensible and formatting cuts out part of the writing.
For all parameters listed in the result section it is not clear whether the authors are refering to the whole number of horses or only to those who survived surgery (which should be the case, since the former would not make sense).
Discussion is very vague and not actually discussing the results. From line the discussion diverges on whether it is an option to perform small intestinal resection or not in general.
Author Response
REVIEWER 4:
R4.1 Table 1 : would make more sense to call it : duration of symptoms.
Changed, see text
R4.2Please adjust the time list in the table in a more logical manner.
Changed, see text
R4.3 Please be consistent throughout the manuscript with the number expressed as numeric and not as spelled in words.
Changed, see text
R4.4 Section on reasons for euthanasia starting in line 133 can be deleted.
Changed, see text
R4.5 I would suggest the result section to consist of less text description and more tables.
Changed, see text
R4.6 In your statistics I am missing torsion of the large colon- but in the section line 159 you refer to strangulating lesions. If you had both types of volvolus of the large colon in your case load, you need to separate them into non-strangulating volvolus of the large colon and large colon torsion, sometimes also referred to as volvolus and indicate the percentages of these also separatly in the tables.
R4.6 a paragraph on strangulating/nonstrangulating lesions has been added in the results
R4.7 Please always spell out abbreviations in the table legend.
Changed, see text
R4.8 Table 5 is incomprehensible and formatting cuts out part of the writing.
Changed, see text
R4.9 For all parameters listed in the result section it is not clear whether the authors are refering to the whole number of horses or only to those who survived surgery (which should be the case, since the former would not make sense).
Changed. Number of horses has been specified
R4.10 Discussion is very vague and not actually discussing the results. From line the discussion diverges on whether it is an option to perform small intestinal resection or not in general.
The whole discussion has been revised

Round 2
Reviewer 1 Report
Manuscript has been appropriately edited.
Author Response
Dear Editor,
We would like to thank the editor and reviewers, we accepted all recommended revisions.
Please see the attachment
